# Breath Analysis: A Systematic Review of Volatile Organic Compounds (VOCs) in Diagnostic and Therapeutic Management of Pleural Mesothelioma

**DOI:** 10.3390/cancers11060831

**Published:** 2019-06-14

**Authors:** Annamaria Catino, Gianluigi de Gennaro, Alessia Di Gilio, Laura Facchini, Domenico Galetta, Jolanda Palmisani, Francesca Porcelli, Niccolò Varesano

**Affiliations:** 1Thoracic Oncology Unit, Clinical Cancer Centre "Giovanni Paolo II", 70124 Bari, Italy; annamaria.catino@gmail.com (A.C.); galetta@oncologico.bari.it (D.G.); nicco.varesano@gmail.com (N.V.); 2Department of Biology, University of Bari, 70125 Bari, Italy; gianluigi.degennaro@uniba.it (G.d.G.); laura.facchini@uniba.it (L.F.); jolanda.palmisani@uniba.it (J.P.); francesca.porcelli@uniba.it (F.P.)

**Keywords:** breath analysis, malignant pleural mesothelioma (MPM), volatile organic compounds (VOCs)

## Abstract

Malignant pleural mesothelioma (MPM) is a rare neoplasm related to asbestos exposure and with high mortality rate. The management of patients with MPM is complex and controversial, particularly with regard to early diagnosis. In the last few years, breath analysis has been greatly implemented with this aim. In this review the strengths of breath analysis and preliminary results in searching breath biomarkers of MPM are highlighted and discussed, respectively. Through a systematic electronic literature search, collecting papers published from 2000 until December 2018, fifteen relevant scientific papers were selected. All papers considered were prospective, comparative, observational case–control studies although every single one pilot and based on a relatively small number of samples. The identification of diagnostic VOCs pattern, through breath sample characterization and the statistical data treatment, allows to obtain a strategic information for clinical diagnostics. To date the collected data provide just preliminary information and, despite the promising results and diagnostic accuracy, conclusions cannot be generalized due to the limited number of individuals included in each cohort study. Furthermore none of studies was externally validated, although validation process is a necessary step towards clinical implementation. Breathomics-based biomarker approach should be further explored to confirm and validate preliminary findings and to evaluate its potential role in monitoring the therapeutic response.

## 1. Introduction

### 1.1. Mesothelioma and Asbestos-Related Diseases

Malignant pleural mesothelioma (MPM) is a rare neoplasm associated with asbestos exposure. Its incidence has been steadily increasing worldwide, although in Western countries a stabilization has been observed in the last decade [1,2,3,4]. The most frequent histotype is the epithelioid variant, while the sarcomatoid subtype is characterized by the worst prognosis and the biphasic variant consists of a combination of the previous ones in different proportions [5]. Asbestos includes six of the 400 mineral fibers detectable in the environment: actinolite, amosite, anthophyllite, chrisotyle, crocidolite and tremolite. They all are regulated for commercial use although all mineral fibers are recognized as probable human carcinogens and the potential health hazards due to the inhalation exposure are well-known [6]. During the last decades, mostly in Europe, the production and the commercial use of asbestos has been restricted or forbidden [7,8,9] resulting in a reduction of the occupational risk, whereas in developing countries its use is still widespread [10,11]. Therefore, due to the very long latency time between inhalation exposure and mesothelioma development, the global incidence of malignant pleural mesothelioma is expected to rise in the next decade [12,13,14]. Moreover, MPM incidence has been attributed to other forms of environmental exposure, although less investigated [1,15,16]. The incidence of the disease is higher in men, while women show a better prognosis; however, a role of circulating estrogens together with ERbeta receptors (in epithelioid subtypes) has been suggested to independently correlate to gender differences [17,18]. Also genetic factors, such as germline variants of BAP-1 Tumor suppressor gene, have been hypothesized to play a role as high-risk factors in the susceptibility to develop malignant mesothelioma in asbestos-exposed subjects [19,20,21]. The carcinogenesis caused by the inhalation of asbestos fibers relies on a local inflammatory condition characterized by the production of cytokine and reactive oxygen species (ROS); furthermore, the asbestos fibers damage the mitotic spindle formation, causing a chromosomal breakage [19].

The inhaled asbestos fibers lead to oxidative stress and stimulate a protracted immune reaction at the pleural sites, thus biomarkers expressing the inflammatory and oxidative status have been deeply investigated [21,22,23]. Serum mesothelin has been identified as a biomarker for the detection of pleural mesothelioma but with limited results [24] while other researchers have investigated the role of serum levels of High Mobility Group Box 1 protein (HMGB1) [25], ostheopontin [26], proteomics-based approaches [27], fibulin-3 and microRNAs [26,28]. More recently, Guarrera et al. [29] have investigated the blood DNA methilation profile, suggesting that early changes correlated to the carcinogenesis could have a role as a biomarker to identify mesothelioma patients and to better estimate the risk in asbestos-exposed subjects.

Overall, to date, a reliable and validated serum biomarker able to identify the subjects when at the higher risk condition of developing pleural mesothelioma is not available. The pathological diagnosis of mesothelioma is challenging, especially in case of small biopsies, due to the difficult differentiation from a typical mesothelial hyperplasia or organizing pleuritis [30,31]; furthermore, the accurate differential diagnosis between some subtypes and other malignant neoplasms often makes the use of a immunohistochemistry panel necessary [30]. The dismal prognosis of MPM is mainly due to the late diagnosis in advanced stage; the standard first-line combination therapy with pemetrexed and a platinum analog allows a mean overall survival and expectation of life of about 12 months [32] while the optimal treatment in second-line setting has not yet been established [33].

The lack of therapeutic options represents a very crucial issue that, however, has prompted the research; more recently insights for innovative treatments such as immunotherapy have been suggested by some clinical studies [34,35,36]. Unfortunately, the early diagnosis of this neoplasm is very challenging due to the limitations of the imaging techniques and the need to carry out invasive diagnostic procedures to discriminate benign conditions from uncertain and/or neoplastic pleural lesions. Hence a foremost and unmet need is to identify a biomarker correlated to the risk of developing the disease, often preceded by pleural plaques and generally characterized by a very long latency time [10,11,12,13,14,15,16]. A non-invasive and reliable method could be useful to better follow, from the therapeutic point of view, subjects at risk due to a previous asbestos exposure in order to facilitate an early diagnosis so improving the treatment and the clinical outcome.

### 1.2. Breath Analysis As Clinical Diagnostic and Disease Monitoring Tools 

In the last few years, breath analysis has been greatly implemented as clinical diagnostic and disease monitoring tool. In the following subsections the description of human breath composition and characteristics, an overview of the analytical methods for breath sampling and analysis as well as a full explanation of the methodological approach applied for review writing are reported.

#### 1.2.1. Breath Composition and Characteristics

Currently physical, biochemical and molecular biological methods, mostly focused on blood and urine analysis, can be considered as widespread routine methods used for medical monitoring and clinical diagnosis. Diagnostics based on breath analysis is much less developed and not yet widely used in clinical practice, even if it is one of the most desirable non-invasive procedures.

Nowadays the breath test is routinely used as a diagnostic tool in the diagnosis of Helicobacter pylori infection through the carbon dioxide monitoring [37] and in the detection of airway inflammatory conditions by the fractional exhaled nitric oxide (FeNO) level monitoring [38,39]. Moreover, the breath test is currently used for the estimation of ethanol and acetaldehyde concentrations in blood (alcohol test) [40]. In the last decade researchers worldwide have developed a new generation of breath tests for the detection of acute and/or chronic diseases, based on the monitoring of endogenous volatile organic compounds (VOCs) in exhaled breath and on the analysis of various cytochines, chemokines and proteins in Exhaled Breath Condensate (EBC) [41,42]. Exhaled breath, indeed, consists of a gaseous phase, containing both inorganic (i.e., nitrogen, oxygen, carbon dioxide, inert gases) and organic species (e.g., VOCs), and a liquid phase containing water vapor and proteins, the EBC. Starting from the assumption that VOCs are conveyed to lungs through the blood system and are exhaled as a result of alveolar gas exchange mechanism, VOCs detection and identification in breath samples has attracted, over the years, the interest of the scientific community and the development VOCs determination-based breath test has been promoted. Therefore, changes in health status of a person and thus in cellular metabolism results in changes of VOCs profile in human breath. Among VOCs, exogenous compounds enter human body via inhalation and skin absorption whereas endogenous ones are generated by biochemical processes such as oxidative stress and fat metabolism. Both classes of VOCs are then catabolized through the cytochrome P450 enzymes and their presence or level changes in breath could be affected by body’s activity [43,44]. Although the knowledge on the composition of human breath in terms of VOCs has been improved in the last decade, the biochemical background of many compounds appearing in exhaled breath remains not completely known [45,46,47]. The monitoring of VOCs in gaseous exhaled breath could represent a new frontier in medical early diagnosis of cancer diseases because tumor growth is mainly accompanied by gene and/or protein changes that may lead to peroxidation of the cell membrane species and, hence, to the release of VOCs. These VOCs can be subsequently detected either directly from the headspace of cancer cells tissues or through body fluids, e.g., exhaled breath [38]. 

Obviously, compared to investigations on human fluids such as blood, urine and stools, the breath sampling and analysis is preferable because is considered a non-invasive approach and avoids potentially infectious waste. In addition, the availability of the samples is essentially limitless and the measurement and detection of volatile compounds in a gaseous matrix is much simpler than in a more complex biologic matrix like the blood [48]. 

Although breath analysis has been performed for some decades, it is still a young field of research. The experimental results are interesting and open up fascinating possibilities of application in cancer research. Besides the several advantages (i.e., non-invasive, simple, fast, risk-free for both patients and medical staff), there are still some limitations such as the lack of standardized analytical methods and not exhaustive knowledge of metabolic processes responsible for the release of the molecules and possible markers of a specific diseases. Sample collection and pre-treatment are methodological key steps because most compounds concentration in the exhaled breath fall in the nmol/L to pmol/L (ppbv to pptv) range.

The composition of exhaled breath could provide valuable information about biochemical processes in the body and offer the rationale for not-invasive diagnostics of cancer and more specifically of pleural neoplasms [38,49,50,51].

#### 1.2.2. Breath Analysis: Sampling and Analytical Methods

Nowadays several analytical methodologies and technologies are applied and used for breath composition determination [52,53]. The gold standard analytical technique for breath analysis is gas chromatography-mass spectrometry (GC-MS) usually combined with thermal desorption (TD-GC/MS) or with solid-phase micro extraction (SPME). The breath is collected directly onto suitable adsorbent materials (i.e., Carbograph, Tenax) or preliminarily inside bags or canisters before being transferred onto adsorbent materials. Polymeric bags are made of inert materials such as Nalophan^®^, Teflon and Tedlar^®^, sometimes covered with outer layers of black Tedlar to block UV rays that may cause compounds’ degradation [54,55]. They are widely used due to their low cost, easy use and potential duration. Stainless steel inert canisters are generally preferred, compared to bags, as their use allows one to avoid light degradation and potential sample contamination or losses due to adsorption and diffusion processes onto/through materials. However, canisters are expensive and need to be cleaned before sampling using specialized cleaning equipment [56,57]. On the basis of limitations described above, systems able to collect breath sample directly on adsorbent materials have been recently developed and are actually recognized as the best approach because sampling steps (e.g., the sample transfer from bags/canisters to adsorbent cartridges is skipped) and potential losses or contaminations are reduced. Once VOCs are collected onto specific cartridges, they are thermally desorbed and separated over a heated GC-column based upon their physical and chemical properties [58]. After the separation, the VOCs are ionized and fragmented in the MS [38,43,59,60,61]. GC-MS allows both identification and quantification of individual compounds with very high sensitivity but the analysis is expensive and requires expert staff and relatively long operational time. On the contrary, selected ion flow tube-mass spectrometry (SIFT-MS), Proton transfer reaction-mass spectrometry (PTR-MS) and ion molecule reaction-mass spectrometry (IMR-MS) allow real-time and on-line measurements of VOCs in breath samples [62]. VOCs are chemically ionized by well-defined reagent ions (i.e., H_3_O^+^, NO^+^ or O^2+^) and transformed in characteristic product ions allowing both detection and quantification. Although PTR-MS is a more sensitive approach compared with SIFT-MS, the latter allows the identification of each compounds in the sample matrix on the basis of the corresponding *m*/*z* value. Moreover, PTR-MS cannot identify substances or differentiate between molecules with the same molecular mass [51,63,64,65,66,67]. Finally, both SIFT-MS and PTR-MS provide real time measurements but GC-MS guarantee the highest sensitivity. 

Another analytical technique that can be used for breath analysis, is ion mobility spectrometry (IMS). The IMS is an analytical technique able to characterize VOCs in breath samples using gas-phase mobility of ions in weak electric fields. Ions mobility depends on the size, mass and shape. IMS equipment consists of a central component, the drift tube, where ion formation and characterization occur, and other components that support the measurement made with the drift tube. The gaseous molecules are firstly ionized, typically by a low energetic radioactive source, and then separated under the influence of a counter gas. In high-level complexity matrices, IMS is often combined with GC multicapillary columns (MCC) in order to efficiently separate VOCs before entering the ionization chamber and drift tube. VOCs are therefore characterized by drift and retention times [62,68,69,70,71,72,73]. Similarly to SIFT-MS and PTR-MS, IMS allows online sampling guarantying a fast and low cost response. However, its sensitivity is lower than that guaranteed by GC-MS although nowadays, in order to improve sensitivity, innovative systems using alternating electrical fields and based on the differential mobility spectroscopy (DMS) have been developed. In order to complete the state-of-the-art on MS-based techniques applied to breath analysis, the ion molecule reaction mass spectrometry (IMR-MS) technique should also be mentioned. IMR-MS instrumentation allow several VOCs to be detected in breath samples at level of ppb without any pre-concentration step in a similar way as PTR-MS. The strength is the ionization process offering the possibility to reduce fragmentation caused by high energy electron impact ionization. IMR-MS approach has been recently applied for VOCs detection in breath samples resulting, in combination with statistical treatments, in an improved diagnostic accuracy of breath analysis for the detection of pancreatic ductal adenocarcinoma [74] and liver diseases such as alcoholic fatty liver disease (AFLD) and non-AFLD (NAFLD) and cirrhosis [75,76]. 

Over the last years, as demonstrated by several recently published studies, sensors have been widely used for pattern recognition and clinical application because they are cheap, easy to handle and small in size. Clinical studies involving sensors highlighted the potentialities of this technology for the identification of specific patterns of markers, avoiding potential interferences. Among them, the electronic nose (e-nose) is a device consisting of an array of chemical sensors with partial specificity developed with the purpose of miming the mammalian olfactory apparatus. Although e-nose technology is not able to perform a qualitative analysis of the sample and therefore to identify individual VOCs, the potentialities of its application in breath analysis are well documented and several studies showed its capability to discriminate among various volatile compounds profiles providing, as an output, a characteristic breath signature [77]. However, it is important to underline that, despite the potentialities in clinical application, sensors used for breath analysis are still affected by poor sensitivity [78,79]. 

### 1.3. Goals of Systematic Review

The main goal of the present review is to provide a state-of-the-art about the use of breath analysis for the management and diagnosis of asbestos-related and MPM diseases. The attention of the authors has been focused on MPM because it is a rare and very aggressive disease with high mortality due to late diagnosis. The current methodological approaches for diagnosis are expensive, invasive and long time consuming and thus, nowadays, a simple and non-invasive tool to assess biomarkers involved in the pathogenesis of asbestos-related and MPM diseases is strongly requested. Several papers in literature showed that asbestos fibers linked to high iron concentrations could directly induce oxidative stress and generate reactive oxygen species (ROS) and nitrogen species (RNS), which determine lipid peroxidation with consequent formation of saturated hydrocarbons and aldehydes [80,81,82,83,84]. Therefore, oxidative stress and inflammation linked to asbestos-related diseases could determine the release of oxidative damage markers, such as VOCs, in breath. These promising findings, therefore, allow the breath analysis to be eligible for the early diagnosis of asbestos-related and MPM diseases. 

### 1.4. Methodological Approach for Review Writing

#### 1.4.1. Literature Search

A systematic electronic literature search was made and only scientific papers in English language published on peer-reviewed journals and based on the studies involving human beings from 2000 until December 2018 were considered. The publications listed in the electronic databases such as United States National Library of Medicine database (Medline–PubMed), NIH U.S. National library of medicine, Scopus and Google scholar were deeply examined by the authors in order to select the most suitable ones for the review purposes. The search process was completed by further scanning of the reference lists obtained from retrieved articles in order to identify additional relevant papers. The selection was then cross-checked and any duplicate and not original articles such as editorials, abstracts and reviews were excluded.

The search terms ‘metabolomic’, ‘Volatile Organic Compounds (VOCs)’, ‘exhaled breath analysis’, ‘asbestosis’ and ‘mesothelioma’ were cross-checked and the risk of bias in individual studies was eliminated by selecting only papers dealing with exhaled breath analysis excluding diagnostic studies on other body fluids. All the selected studies on VOCs in asbestosis and MPM were included regardless of the outcome, eligibility criteria for medical trials and aim (screening or follow up monitoring) of the study.

#### 1.4.2. Search Strategy and Literature Selection 

Figure 1 shows the process of papers selection according to the Preferred Reporting Items for Systematic Reviews and Meta-Analyses (PRISMA) standards and guidelines [85]. In a preliminary search taking into account single keywords and the aforementioned electronic databases, more than 76000 papers were found. After analysis of the papers content and considering the keywords ‘asbestosis exhaled breath analysis’ and ‘mesothelioma exhaled breath analysis’, 1089 and 1368 papers were found, respectively. The check of reference lists highlighted further 4 interesting studies. Twenty seven papers were deeply screened after elimination of duplicates and this procedure resulted in the exclusion of further 12 references because not in line with the review’s objectives. Finally, 15 papers were considered eligible for a full-text analysis and included in the present systematic review. 

#### 1.4.3. Data Collection and Analysis

The studies included in the present review are prospective, comparative, observational case–control studies although based on a relatively small number of samples. A schematic overview on the experimental design, the applied methodology and population characteristics is shown in Table 1. The qualitative and quantitative information of the selected papers were drawn by the authors according to the recommendations reported in Cochrane Collaboration for diagnostic research [86].

## 2. Results and Discussion

### 2.1. Gas Chromatography and Liquid Chromatography Coupled to Mass Spectrometry (GC/MS, LC/MS)

Gas chromatography coupled to mass spectrometry (GC/MS) remains the gold standard analytical technique as it is very sensitive and allows both identification and quantification of individual compounds. The drawbacks of this analytical technique are long operation time, costs and requirement for expert operator staff. GC/MS technique is usually combined with thermal desorption (TD) or solid-phase micro extraction (SPME). Several research groups investigated the efficiency of GC/MS in diagnosing asbestosis and pleural mesothelioma performing tests both on EBC and on gaseous matrices. Liquid chromatography (LC) is, instead, specifically used for the analysis of EBC. A critical discussion of the experimental results obtained by the application of chromatographic methodologies for human breath analysis is reported as follows (Table 1).

De Gennaro et al. [95] set up and validated a TD-GC/MS technique in order to perform early discrimination among 13 patients affected by MPM, 13 subjects without MPM but with long-term professional exposure to asbestos (EXP) and 13 healthy controls (HC). Patients were asked to breath tidally for 5 min through a mouthpiece connected to a three-way non-rebreathing valve with an inspiratory VOC-filter at the inlet side. After a deep inspiration through the nose, the patient exhaled a single capacity volume. Samples were collected inside 5 L Tedlar bags and then transferred on three adsorbent beds cartridges (e.g., Carboxen 1003, Carbopack B, CarbopackY, Sigma Aldrich, Merck KGaA, Darmstad, Germany) and analyzed by using a thermal desorber (Unity 1™, Markes International Ltd., Llantrisant, United Kingdom) coupled with a gas chromatograph (Agilent GC-6890 PLUS, Santa Clara CA, USA) and a mass selective detector (Agilent MS-5973 N, Santa Clara, CA, USA). Concentration values were examined by the application of univariate (ANOVA) and multivariate statistical treatments (i.e., PCA, DFA and CPANN) and showed that cyclopentane and cyclohexane were the dominant ‘variables’ able to discriminate among the three groups. Cyclohexane allowed to differentiate the MPM group from the groups EXP and HC while cyclopentane was useful for discrimination between EXP and the groups MPM and HC. Lamote et al. [87] carried out a multicenter, cross-sectional case-control study exploiting GC/MS potentialities for the identification of VOCs in MPM patients exhaled breath and benefiting of eNose performances for screening activity. Fourteen HC, 19 asymptomatic former EXP individuals, 15 patients with benign asbestos-related diseases (ARD) and 14 MPM patients were involved in the study. Similarly to de Gennaro et al. patients were asked to exhale maximally after 5 min of tidal breathing. Samples were collected inside 10 L Tedlar bags using a two-way non-rebreathing valve (Hans Rudolph 2700, Hans Rudolph, Kansas City, MO, USA) provided with a inspiratory VOC-filter at the inlet side (A2, North Safety, Middelburg, The Netherlands). Collected samples were then transferred on adsorbent cartridges (Tenax^®^GR 35/60 mesh, Markes International Ltd., Llantrisant, United Kingdom) and analyzed using a thermal desorption system (Markes International Ltd., Llantrisant, United Kingdom) coupled to a GC/MS (Thermo Finnigan, Austin, TX, USA). Variables expressed as mean or median values were prior treated using R (v3.3.1) studio interface, categorical variables were compared using a Pearson Chi2-test and reported as ratios while continuous variables normality was checked by a Shapiro-Wilk test. Due to the high number of variables and the limited number of breath samples, a penalized logistic regression using the least absolute shrinkage and selection operator (e.g., lasso, *glmnet* R-package (v2.0-2)) was applied. Using the predicted outcomes of all patients, the ROC curve was derived (ROCR R-package (v1.0-7, Environmental Research Group, King’s College, London, UK) and, as a result, the area under the curve was calculated (AUCROC, 95% confidence intervals) and sensitivity, specificity, positive (PPV) and negative predictive values (NPV) as well as diagnostic accuracy of the final model was estimated. The statistical treatment highlighted the possibility to discriminate between EXP and MPM patients with 97% accuracy and that specific VOCs such as diethyl ether, limonene, nonanal, methylcyclopentane and cyclohexane were useful for discrimination. MPM patients were identified with 94% of accuracy and sensitivity, specificity, positive and negative predictive values were 100%, 91%, 82%, 100% respectively.

Since 2008, four different studies were specifically carried out on EBC. Pelclova et al. [99] tested the level of oxidative stress marker in EBC of 92 former asbestos workers and 46 HC subjects without occupational exposure to asbestos but with a lifestyle characterized by factors potentially influencing oxidative stress EBC samples were collected using the EcoScreen ( Erich Jaeger GmbH, Haechberg, Germany). Each subject breathed through the collection kit for 15 min and the 2 mL of collected EBC was immediately frozen at –80 °C. 8-isoprostane in EBC was analyzed after immunoaffinity separation using liquid chromatography-electrospray ionization-mass spectrometry (LC-ESI-MS) in multiple reaction monitoring mode (MRM). The experimental results were statistically treated by applying Student’s *t*-test, *F*-test, ANOVA, χ^2^ and linear regression and showed 8-isoprostane levels in asbestos-exposed subjects higher than healthy controls.

Moreover, Syslová et al. [98] published two consecutive papers focused on analysis of EBC collected from patients affected by occupational lung diseases. In 2009 the authors pointed out a preliminary study to develop a sensitive assay method for a parallel, rapid and precise determination of the most prominent oxidative stress biomarkers: 8-*iso*-prostaglandin F, malondialdehyde and 4-hydroxynonenal. EBC were collected by means of EcoScreen condenser (Erich Jaeger GmbH, Haechberg, Germany) from healthy subjects and patients affected from asbestosis, pleural hyalinosis or silicosis, i.e., occupational lung diseases caused by fibrogenic dusts. Patients performed a tidal breathing for 5–10 min through a mouthpiece connected to the condenser and the collected volume of EBC samples ranged from 1–2 mL. Samples were then analyzed by SPME coupled with liquid chromatography-electron spray ionization/tandem mass spectrometry (LC-ESI-MS/MS). The comparison of biomarkers concentration levels allowed to observe that a significant difference between the two groups existed. A further study carried out by the same authors in 2010 was principally focused on the development of a selective and sensitive method for the quantification of 8-*iso*-prostaglandin F2*α* (8-*iso*-PGF2*α*), *o*-tyrosine (*o*-Tyr) and 8-hydroxy-2-deoxyguanosine (8-OHdG) in EBC as they were recognized as significant biomarkers of oxidative stress in vivo [96]. The method was then tested on EBC samples collected by means of a commercial EcoScreen (Erich Jaeger GmbH, Haechberg, Germany) from 10 patients with occupational lung diseases, either silica- or asbestos-induced disorders, and 10 subjects without any occupational exposure to fibro genic dusts. EBC analysis was carried out by LC-ESI-MS/MS and the results were treated by Student’s *t*-test (Statistica, version 6.0, Dell Software, Round Rock, TX, USA). A statistical significance of *o*-Tyr and 8-OHdG levels between the two groups was observed, not for 8-*iso*-PGF2*α*. However, due to the small number of subjects involved in experimentation, general conclusions cannot be drawn.

### 2.2. Ion Mobility Spectrometry 

Lamote et al. carried out three IMS-based experimental studies in 2014, 2016 and 2017 respectively [88,90,91]. The multicenter cross-sectional and case-control studies published in 2014 and 2016 aimed to explore the possibility to use IMS to discriminate between patients affected by MPM asbestos exposed workers (EXP) and HC involving 20 vs. 23 MPM patients, 10 vs 22 asymptomatic former asbestos-exposed workers and 10 vs 21 HC respectively [90,91]. The goal of the last study, published in 2017, was to validate earlier findings on a larger population and discriminate by VOCs analysis between patients affected by MPM and lung cancer (LC) [88]. For this purpose, 52 MPM patients, 56 primary lung cancer patients, 70 subjects with benign non-asbestos related lung diseases, 41 subjects with benign asbestos related diseases, 59 subjects with asymptomatic former asbestos exposure and 52 HC were enrolled. In all the aforementioned studies by Lamote et al., breath samples were collected and analyzed by using a BioScout device (B&S Analytik, Dortmund, Germany) consisting of a ion mobility spectrometer coupled with a multicapillary column (MCC) and connected to a SpiroScout ultrasound-controlled breath sampler (Ganshorn Medizin Electronic, Niederlauer, Germany) by a sample loop. By capno-volumetry, the SpiroScout detects the CO_2_-levels in exhaled breath and starts the breath sampling when a plateau in CO_2_-levels is reached, indicating that the alveolar air is sampled. VOCs peaks were selected analyzing the 2D-chromatograms with VisualNow software (v3.2 in the first study and v3.7 in the others; B&S Analytik, Dortmund, Germany) and subsequently normalized to the reactant ion peak (RIP). After a visual inspection of all samples, VOCs were manually selected and then the list of VOC-peak intensities was obtained and deeply studied. To remove the impact of environmental chemical confounders, the alveolar gradient was calculated by subtracting the standardized peak intensity in the background air samples. Due to the high number of variables and the rather low number of samples a logistic regression method was used to search for peaks that have the most discriminative power (R LASSO logistic regression, R Foundation for Statistical Computing, Vienna, Austria). Using the outcomes and considering the number of times a single VOC was selected by the lasso regressions, the ROC curve was derived and sensitivity, specificity, positive and negative predictive values (PPV, NPV) and diagnostic accuracy were estimated (95% confidence intervals). In the two most recent studies Fisher’s exact test, Kolmogorov-Smirnov test, ANOVA or Kruskal-Wallis test were also applied [88,90]. 

In the first study carried out in 2014 a discrimination between MPM patients, asbestos-exposed and non-exposed controls with 85% sensitivity (64–96%) and 90% specificity (71–98%) was observed. The AUCROC was 0.92 and the PPV and NPV were 90% (69–98%) and 86% (66–96%) respectively. The results obtained highlighted that age and the VOCs P5, P3, P83, P1 and P67 play a key role as discriminators [91]. The methodological approach developed later (e.g., Lamote et al., 2016 [90]) allowed the discrimination between MPM patients and controls with 87% sensitivity, 70% specificity and respective positive and negative predictive values equal to 61% and 91%. The overall accuracy was 76% and the area under the ROC-curve was 0.81. Asymptomatic former asbestos workers (AEx), instead, were discriminated from MPM patients with 87% sensitivity, 86% specificity and respective positive and negative predictive values equal to 87% and 86%. The overall accuracy was 87% with an area under the ROC-curve of 0.86. In this study the VOCs with a better discriminative power were P3, P5, P50 and P71.

The comparison made in 2017, including in the study also lung cancer patients, highlighted the possibility to discriminate between MPM patients and HC, AEx, ARD, benign non-asbestos-related lung diseases (BLD) and LC patients with 65%, 88%, 82%, 80% and 72% accuracy, respectively [88]. Including AEx and ARD patients in a unique group, sensitivity and negative predictive value (NPV) increased reaching 94% and 96% respectively. The most discriminating VOCs were P1, P3, P7, P9, P21, and P26. However, the uncertainty of the chemical identification did not allow to make a correct and complete comparison between the obtained results. 

The potential of MCC-IMS for diagnostic purposes was also investigated by Cakir et al. [92]. The main objective of the study was to verify the potentiality of the methodology for discrimination between 25 MPM and 12 HC. Box and Whisker plot construction as well as a decision tree were used allowing α-pinene and 4-ethyltoluol to be identified as the most useful VOCs for discrimination with a sensitivity of 96%, a specificity of 50%, positive and negative predicted values of 80% and 86%, respectively. Although the aforementioned VOCs were identified by Cakir et al. as most discrimination peaks as characterized by lower p-values, it is important to highlight that α-pinene has exogen origin and that its presence in human breath samples could be related to potential contamination and/or metabolic disorder in asbestosis- related diseases patients. 

### 2.3. Sensor Technology 

Breath samples can be investigated in terms of composition by using sensor technology, an approach based on pattern recognition. Sometimes it is possible to use analyzer equipped with specific sensor able to recognize the presence of a single molecule or a pattern of compounds. Commercial devices based on sensor technology were also tested on samples collected by MPM patients in order to develop a reliable and fast approach to diagnose the disease.

An e-nose made of a carbon polymer array (Cyranose 320; Smiths Detection, Pasadena, CA, USA) was used in three different studies. Chapman et al. aimed to discriminate between 20 MPM patients, 13 subjects affected by pleural disease and 42 HC [93]. Dragonieri et al. used an e-nose in order to discriminate among 13 MPM patients, 13 subjects asbestos-exposed and 13 HC [94]. Few years later Lamote K. et al. [87] compared the results obtained from the analysis of breath samples collected from 14 MPM patients, 15 subjects with benign asbestos related diseases, 19 asbestos exposed and 16 HC applying the gold standard technique (GC-MS) and using four different e-noses: Cyranose C320, Tor Vergata eNose, Common Invent eNose and Owlstone Lonestar. A preliminary statistical treatment of the obtained data was performed applying Principal Component Analysis (PCA). Moreover, Savitzky–Golay filtering was applied to process the sensor response data and baseline corrections were applied to improve signal-to-noise ratio. PCA factors were used to perform a linear canonical discrimination analysis for the construction of a pattern recognition algorithm. A cross validation value (%) and the Mahalanobis distance (M-distance) between group mean values, in units of standard deviation (SD), were calculated. Breath samples of 10 MPM subjects were used to create the training set and then the model was tested on the other 10 MPM obtaining an accuracy in discrimination between MPM and HC samples equal to 95%. Patients with MPM, ARDs and control subjects were correctly identified in 88% of cases. E-nose raw data from Dragonieri et al. were analyzed by SPSS software version 16.0 (SPSS Inc., Chicago, IL, USA) [94]. Data were reduced to a set of principal components and an independent t-test was used to select the more discriminating principal components. The selected principal components were used for linear canonical discriminating analysis (CDA) with the purpose to create a model able to maximize the distance between sample classes and minimize the within-sample class distances. The probability of a positive diagnosis was calculated on basis of the canonical discriminating function and used to create a receiver operator curve (ROC-curve) (95% confidence limits). Performing a three-way classification of MPM patients, asbestos exposed and HC, the cross validated accuracy percentage (CVA%) equal to 79.5% (*p* = 0.001) and the area under the ROC-curve of 0.885 were obtained. 

Lamote K. et al. used e-nose technology to validate the results obtained with GC/MS and to discriminate MPM patients from AEx+ARD subjects [87]. The sensitivity, specificity, positive and negative predictive values were 82%, 55%, 82%, 55%, respectively. The raw eNose data were processed by PCA, categorical variables were compared using a Pearson Chi^2^-test while, for continuous variables, normality was checked by a Shapiro-Wilk test.

Lehtonen et al. and Chow et al. focused their attention on EBC composition [97,100]. More specifically, Lehtonen et al., investigated whether exhaled NO or leukotriene B4 and 8-isoprostane (inflammatory markers) in EBC could be used to assess inflammation in asbestosis. EBC samples of 15 patients with asbestosis and 15 HC were collected during 15 min of tidal breathing using Ecoscreen condenser (Erich Jaeger GmbH, Haechberg, Germany) and stored at –70 °C. NO was monitored by means of Sievers NOA 280 analyser (Sievers Instruments, Boulder, CO, USA) while LTB4 and 8-isoprostane concentrations by immunoassay with a detection limit of 1.95 pg/mL (Cayman Chemical Company, Ann Arbor, MI, USA). SPSS Version 10.1 software was used to treat the data expressed as mean values and all the parameters were normally distributed according to the Kolmogorov-Smirnov test. The experimental results highlighted that the alveolar concentration of NO at highest exhalation flow rate was higher in patients with asbestosis than in healthy controls, while there were no differences in bronchial NO fraction. The concentrations of LTB4 and 8-isoprostane in exhaled breath condensate were also higher in patients with asbestosis than in healthy controls. 

Chow et al., aimed to assess lung oxidative stress and inflammation in vivo in subjects with asbestos-related disorders. Eighteen patients with asbestosis, 26 with pleural plaques, 16 with diffuse pleural thickening (DPT) and 26 HC were involved in the study [97]. EBC was collected using Ecoscreen (Erich Jaeger GmbH, Haechberg, Germany) and stored at −80 °C. EBC acidity was measured immediately after defrosting frozen EBC samples with a pH sensor probe (IQ125 MiniLab Professional pH meter, Merck, KGaA, Darmstad, Germany). Enz-Chek Ultra Amylase kit (Molecular Probes, Invitrogen, Thermo Fischer Scientific, Carlsbad, CA, USA) was used to test the presence of a-amylase in samples while FeNO and exhaled CO were measured online using a rapid-response chemiluminescence NO and CO analyzer (LR 2500 (I), Logan Research, Rochester, UK). Total nitrogen oxides (NOx) were measured after enzymatic reduction of nitrate using a fluorimetric modification of the Greiss reaction and total protein concentration using a Quantipro BCA assay kit (Sigma Aldrich, Sydney, Australia). 8-Isoprostane was measured using a specific enzyme-immunoassay (EIA) kit (Cayman Chemical), validated to obtain a high correlation (0.95) with known amounts of 8-isoprostane. Hydrogen peroxide (H_2_O_2_), instead, was measured spectrophotometrically by horseradish peroxidase-catalyzed oxidation of tetramethylbenzidine. The nitration product of tyrosine, 3-nitrotyrosine, was measured via enzyme immunoassay (EIA, Cayman Chemical). Data, expressed as mean values, were then examined by analysis of variance (ANOVA) and specific statistical tests were used to compare groups. Correlation between biomarkers and lung function parameters was performed using Pearson’s correlation coefficient. The results of the study highlighted that markers of inflammation and oxidative stress are significantly elevated in subjects with asbestosis compared with healthy individuals but not in pleural diseases. Patients with asbestosis showed higher levels of oxidative stress markers in EBC. The concentrations of 8-isoprostane and hydrogen peroxide in patients with asbestosis were higher than those determined in normal controls samples as well as increased EBC total protein and FeNO. EBC pH was lower in subjects with asbestosis compared with subjects with DPT. No significant differences were observed in levels of exhaled carbon monoxide, EBC total nitrogen oxides and 3-nitrotyrosine between different kind of asbestos-related disorders, or between these and healthy controls. Moreover, recently, some studies have pointed out the correlation between fractional exhaled nitric oxide concentration (FeNO) and CO concentration with lung disorders. Taking into account that several factors may affect FENO levels, volunteers was recruited considering as the exclusion criteria the history or present symptoms of asthma, allergy, impaired lung function, recent upper respiratory tract infection, sinusitis and active or passive smoking. The FeNO levels in subjects with asbestosis and pleural plaques resulted higher than in normal controls while no significant differences were reveled in exhaled carbon monoxide [89,101]. 

### 2.4. Canine Scent

Sensor technology, more specifically e-nose, is developed to reproduce natural olfactory perception. Considering the ability of human olfaction and the good performance showed by trained sensors, a completely novel technique was recently tested for the analysis of human breath. Sensitive smelling ability and learning capacity of trained canine scent was used to discriminate different patterns with good preliminary results. To date, however, a specific study has been not performed yet for discrimination of mesothelioma and asbestosis and in general of lung cancer. Only one patient with mesothelioma was included by Amundsen T et al. 2014 in the cohort of 93 patients for which canine olfactory test was performed on urine and breath samples in a double-blinded manner [102]. The test resulted in 99% sensitivity in the discrimination between cancer patients and healthy individuals but not specific comment was reported from authors on the case of mesothelioma. 

Preliminary other studies training dog stocks to recognize and discriminate different breath samples of lung cancer patients and healthy controls were carried out and showed a sensitivity and a specificity greater than 70% and 80%, respectively [103,104].

## 3. Conclusions and Future Perspectives

Although MPM is considered a rare disease, mainly caused by exposure to asbestos fibers, the number of deaths caused by this neoplasm is still increasing worldwide due to its aggressiveness and usually late diagnosis. Therefore, a reliable screening technique for MPM early-diagnosis is needed in order to increase patients’ survival. The studies reported in the present review suggest that breath analysis is a promising technique for this purpose because it could represent a non- invasive, easy to use and a reliable tool. The identification of a distinct mesothelioma-related VOCs profile through breath sample by using both analytical techniques and/or sensor technology and the statistical elaboration of data by specific data mining approaches, could allow for obtaining a sufficient diagnostic power to differentiate among asbestos-exposed subjects, patients affected with pleural mesothelioma and healthy controls. However, to date, the available data provide only preliminary information so, despite the promising results, conclusions from these cohort studies cannot be generalized due to the small number of subjects included. Furthermore, none of these studies have externally validated their findings, which is a necessary step towards clinical implementation. Therefore, further research will be useful in order to confirm the previous findings as well as to refine the VOCs signature by implementing an experimental protocol dedicated to the disease. Moreover, it would be extremely important to deepen the mechanisms underlying the production of VOCs by the cancer cells and the inflamed stromal environment.

Actually, to identify a VOC signature for the malignant pleural mesothelioma could meet the need for screening (especially directed to asbestos-exposed at-risk subjects) through a simple breath collection. Furthermore, the same tool might help to monitor the therapeutic response, as well as to detect a disease, recurrence during the follow up of patients. To this aim, further studies are strongly warranted.

## Figures and Tables

**Figure 1 cancers-11-00831-f001:**
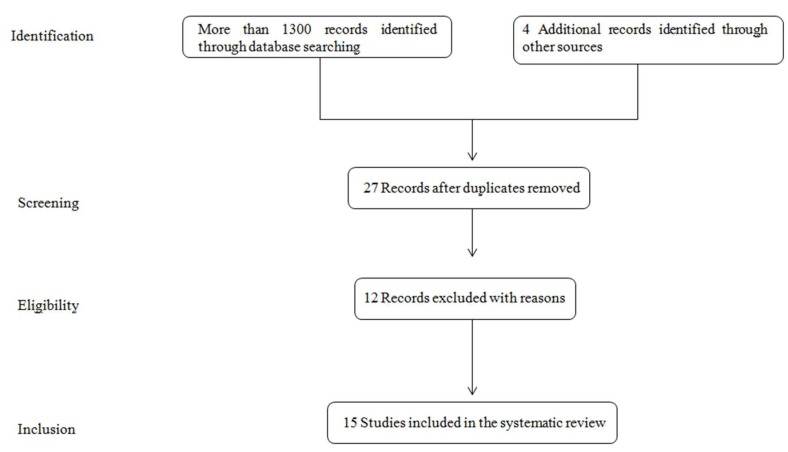
Study selection process.

**Table 1 cancers-11-00831-t001:** Overview of experimental design, applied methodology and population characteristics of selected papers.

First Author, Year [Ref]	Sample Tipology	Analytical Technique	Storage Container	Subjects Involved (Cancer Stage)	Classifier	Results (Discriminating VOCs)	Concentration Range
Lamote K., 2017 [87]	exhaled breath samples	GC/MS e-nose	10 L Tedlar bags	14 MPM, 19 asymptomatic former EXP, 15 ARD, 14 HC	Pearson Chi2-test, Shapiro-Wilk test, logistic regression (lasso)	diethyl ether, limonene, nonanal, methylcyclopentane and cyclohexane	not reported
Lamote K., 2017 [88]	exhaled breath samples	MCC-IMS		52 MPM, 56 LC, 70 BLD, 41 ARD, 59 EXP and 52 HC	Logistic regression (lasso), Fisher’s exact test, Kolmogorov-Smirnov test, ANOVA or Kruskal-Wallis test	P1, P3, P7, P9, P21, and P26	not reported
Karvonen T., 2017 [89]	exhaled breath samples	Sievers NOA 280 chemiluminescence analyser		69 HC adults, 66 HC children, 73 asbestos-exposed and 72 COPD	5 mathematical methods (Tsoukias and George, Pietropaoli, Condorelli, Högman and Meriläinen, and Silkoff) to estimate alveolar and bronchial NO parameters	NO	not reported
Lamote K., 2016 [90]	exhaled breath samples	MCC-IMS		23 MPM, 22 asymptomatic former asbestos workers and 21 HC	Logistic least absolute shrinkage and selection operator (lasso) regression, chi-squared test or Fisher’s exact, Kolmogorov–Smirnov test to assess normality, t-test or analysis of variance (ANOVA), Wilcoxon–Mann–Whitney test or Kruskal–Wallis test	P3, P5, P50 and P71	not reported
Lamote K., 2014b [91]	exhaled breath samples	MCC-IMS		20 MPM patients, 10 asbestos-exposed and 10 HC	Logistic LASSO regression	P5, P3, P83, P1 and P67	not reported
Cakir Y., 2014 [92]	exhaled breath samples	MCC-IMS		25 MPM and 12 HC	Box and Whisker plot and decision tree	4-ethytoluol and alpha pinene	not reported
Chapman E.A., 2012 [93]	exhaled breath samples	e-nose (Cyranose320)	2-L gas impermeable bag	20 MPM (19:stage 2, 1:stage 1b), 13 Pleural disease and 42 HC	PCA, linear canonical discriminant analysis and Mahalanobis distance		not reported
Dragonieri S., 2012 [94]	exhaled breath samples	e-nose (Cyranose 320)	5-L Tedlar bag	13 MPM (7:stage 1b; 1:stage 1a; 3: stage 2; 2: stage 3), 13 asbestos-exposed and 13 HC	PCA and CDA		not reported
de Gennaro G., 2010 [95]	exhaled breath samples	TD-GC-MS	5-L Tedlar bag	13 MPM, 13 EXP and 13 HC	Anova, PCA, DFA, CP-ANN	cyclopentane, cyclohexane	cyclopentane median value: MPM patients 120.42 ng/L vs. asbestosis 605.49 ng/L vs. HC 34.83 ng/L.; cyclohexane median value: MPM patients 251.79 ng/L vs. asbestosis 69.31 ng/L vs. HC 33.08 ng/L.
Syslová K., 2010* [96]	EBC	LC-ESI-MS*/*MS		10 patients occupational lung diseases (either silica or asbestos exposure) and 10 HC	Student’s *t*-test	8-*iso*-PGF2α, o-Tyr and 8-OHdG	Patients vs. HC (median pg ml−1): 8-*iso*-PGF2α 106.1 vs. 86.7; o-Tyr 61.8 vs. 47.5; 8-OHdG 46.5 vs. 14.8.
Chow S., 2009 * [97]	EBC	fluorimetric modification of the Greiss reaction (NOx), Quantipro BCA assay kit (total protein), enzyme-immunoassay (EIA) kit (8-Isoprostane), enzyme immunoassay (EIA) (3-nitrotyrosine) and H_2_O_2_ measured spectrophotometrically		18 Asbestosis, 26 Pleural plaques, 16 diffuse pleural thickening (DPT) and 26 HC	Anova and Pearson’s correlation coefficient	8-isoprostane, leukotrienes B4, C4, D4, and E4, hydrogen peroxide, EBC total protein, fractional FeNO	asbestosis vs. HC: 8-isoprostane (geometric mean (95% CI) 0.51 (0.17-1.51) vs. 0.07 (0.04–0.13) ng/mL); hydrogen peroxide (13.68 (8.63–21.68) vs. 5.89 (3.99–8.69); EBC total protein (17.27 (10.57–28.23) vs. 7.62(5.13–11.34) mg/mL; FeNO (mean +/− SD) (9.67 +/− 3.26 vs. 7.57 +/− 1.89 ppb).
Syslová K., 2009 * [98]	EBC	LC-ESI_MS/MS		20 patients occupational exposure to asbestos/silica dust (for 24 years in average) and 10 HC	Student’s *t*-test	8-*iso*prostaglandin F, malondialdehyde(MDA) and 4-hydroxynonenal (HNE)	asbestosis vs HC: 8-iso (mean 71(66–77) vs. 52 (45–61) pg/mL); MDA (mean 72 (65–86) vs. 45 (36-55) ng/mL); HNE (mean 233 (188–267) vs. 165 (140–186) ng/mL).
Pelclova D., 2008 * [99]	EBC	LC-ESI-MS		92 asbestos-exposed and 46 HC	Student’s *t*-test, F-test, ANOVA and linear regression	8-isoprostane	8-isoprostane asbestos exposed 69.5 ± 6.6 pg/mL vs. HC 47.0 ± 7.8 pg/mL
Lehtonen H., 2007* [100]	EBC	Sievers NOA 280 analyser and immunoassay kit		15 Asbestosis and 15 HC	Mean values	NO, LTB4 and 8-isoprostane	asbestosis vs HC: NO (3.2 (0.4) vs. 2.0 (0.2) ppb); LTB4 (39.5 (6.0) vs 15.4 (2.9) pg/mL);8-isoprostane (33.5 (9.6) vs. 11.9 (2.8) pg/mL).
Sandrini A., 2006 [101]	exhaled breath samples	chemiluminescence NO and CO analyser LR 2500 (I)		56 subjects with asbestos-related disorders and 35 HC	Anova and multiple comparison post hoc test (Scheffe)	NO, CO	FENO: asbestosis (7.9 (6.6–15.7) ppb), pleural plaques (6.3 (5.3–-9) ppb), HC (4.6 (3.5–6) ppb).

* Exhaled Breath Condensate (EBC); MPM = malignant pleural mesothelioma; LC = primary lung cancer; BLD = benign non-asbestos related lung diseases; ARD = benign asbestos related diseases; EXP = asymptomatic former asbestos exposed; HC = healty controls non exposed; CDA = Canonocal Discriminant Analysis; PCA = Principal Component Analysis; DFA = Discriminant function analysis; CP-ANN = Counterpropagation artificial neural networks; COPD = chronic obstructive pulmonary disease.

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
