# Peer review of "Breath Analysis: A Systematic Review of Volatile Organic Compounds (VOCs) in Diagnostic and Therapeutic Management of Pleural Mesothelioma"

_cancers, 2019, doi:10.3390/cancers11060831_

Round 1

Reviewer 1 Report

This is an interesting review that tries to summarise the literature on VOCs as markers for mesothelioma.

The authors summarise the findings of the original papers, but are not critical of the compounds and what they might mean. It is implausible, in my opinion, that alpha-pinene and 4-ethyltoluene are really markers of mesothelioma: this are usually plant derived compounds. Why would they be biomarkers?

When discussing markers of inflammation (NO etc), the controls should be considered. NO is raised in asthma! So the authors should reflect on the choice of controls and the limitations of the findings.

Minor comments

There are a lot of typographic and spelling errors

Table 1 needs to be realigned

I think the methods (search strategy) should precede the results - but this may be appropriate for the journal.

Author Response

Dear reviewer,

we reviewed the paper according to your relevant and useful  suggestions. Moreover, the manuscript has been thoroughly revised in terms of content adding further information and for grammar and spelling by a mother tongue English scientist.

Please read through the manuscript. We answered, point by point, to your comments and reviewed the paper hoping that it will be more acceptable for publication.

REVIEWER #1:

This is an interesting review that tries to summarize the literature on VOCs as markers for mesothelioma.

The authors summarize the findings of the original papers, but are not critical of the compounds and what they might mean. It is implausible, in my opinion, that alpha-pinene and 4-ethyltoluene are really markers of mesothelioma: this are usually plant derived compounds. Why would they be biomarkers?

Response to reviewer: The authors agree with the reviewer’s comment and added in the text a critical and clarifying sentence on the presence of alpha-pinene in human breath samples (Read through lines 322-326, please). To our knowledge, there are no clear evidences concerning the origin and/or metabolic pattern of 4-ethyltoluene. Therefore, in-depth observation about this specie is not provided in the text.

When discussing markers of inflammation (NO etc), the controls should be considered. NO is raised in asthma! So the authors should reflect on the choice of controls and the limitations of the findings.

Response to reviewer: The authors agree with referee's suggestion and they revised the manuscript specifying the potential confounding factors and volunteers recruiting criteria of the considered studies. Read through the line 491-494, please.

Minor comments

There are a lot of typographic and spelling errors

Response to reviewer: The authors thanks referee and the manuscript has been thoroughly revised for grammar and spelling by a mother tongue English scientist

Table 1 needs to be realigned

Response to reviewer: Done

I think the methods (search strategy) should precede the results - but this may be appropriate for the journal.

Response of reviewers: The authors thank the reviewer for the provided suggestion. Although the manuscript template for review let the authors decide about the sections order, the manuscript has been re-organized and the sections “literature search”, “Search Strategy and Literature Selection” and “data collection and analysis” (including Table 1) precede the section “results and discussion”.

Reviewer 2 Report

Manuscript ID: cancers-508959

The paper “Breath analysis: A Systematic Review About Volatile Organic Compounds (VOCs) in Diagnostic and Therapeutic Management of Pleural Mesothelioma” is interesting and well done.

I suggest only 3 adjustments:

-          Line 110 “To date the biochemical background of many compounds appearing in exhaled breath remains not completely known [45].” This reference is dated 2006: I’m sure that the biochemical background is improved in the last decade, despite the persistence of large and basic lacunas. You should include a more recent reference.

-          Lines 150-158:  The paragraph 1.2.2. “Breath analysis: sampling and analytical methods” should include a few information about the Gas analysis by IMR-MS. This instrument can measure several volatile compounds at level of ppb without any preconcentration, in a similar way as PTR/MS It cannot identify substances, but it can work on line and measure several volatile compounds, in a few seconds, at level of ppb, such as ammonia, nitrogen gas, mercaptanes, organic sulphides, acetic acid and other VOC. The researches made with GC/MS can be continued and integrated by using also this kind of instruments.

-          In Table 1 the column “Classifier” reports the statistical elaboration used by different working groups. It is not clear if the reported data “ Logistic regression (lasso), Fisher’s exact test, Kolmogorov-Smirnov test, ANOVA or Kruskal-Wallis test 5 mathematical methods (Tsoukias and George, Pietropaoli, Condorelli, Högman and Meriläinen, and Silkoff) to estimate alveolar and bronchial NO parameters Logistic least absolute shrinkage and selection operator (lasso) regression, chi-squared test or Fisher’s exact, Kolmogorov–Smirnov test to assess normality, t-test or analysis of variance (ANOVA), Wilcoxon–Mann–Whitney test or Kruskal–Wallis test” are related to the all three first working groups (Lamote K., 2017 [87], Karvonen T., 2017 [93] and Lamote K., 2016 [87]) or to only the first one.  Please, make table 1 clearer.

Author Response

Dear reviewer,

we reviewed the paper according to your relevant and useful  suggestions. Moreover, the manuscript has been thoroughly revised in terms of content adding further information and for grammar and spelling by a mother tongue English scientist.

Please read through the manuscript. We answered, point by point, to your comments and reviewed the paper hoping that it will be more acceptable for publication.

I suggest only 3 adjustments:

-          Line 110 “To date the biochemical background of many compounds appearing in exhaled breath remains not completely known [45].” This reference is dated 2006: I’m sure that the biochemical background is improved in the last decade, despite the persistence of large and basic lacunas. You should include a more recent reference.

Response to reviewer: The authors agree with referee's suggestion and they revised the manuscript and updated references. Read through lines 115-117 , please.

-          Lines 150-158:  The paragraph 1.2.2. “Breath analysis: sampling and analytical methods” should include a few information about the Gas analysis by IMR-MS. This instrument can measure several volatile compounds at level of ppb without any preconcentration, in a similar way as PTR/MS. It cannot identify substances, but it can work on line and measure several volatile compounds, in a few seconds, at level of ppb, such as ammonia, nitrogen gas, mercaptanes, organic sulphides, acetic acid and other VOC. The researches made with GC/MS can be continued and integrated by using also this kind of instruments.

Response to reviewer: The authors added in the text of “Breath analysis: sampling and analytical methods” section a brief description of IMR-MS technique with references of studies where this approach has been applied. Read through lines 182-191 , please.

-          In Table 1 the column “Classifier” reports the statistical elaboration used by different working groups. It is not clear if the reported data “ Logistic regression (lasso), Fisher’s exact test, Kolmogorov-Smirnov test, ANOVA or Kruskal-Wallis test 5 mathematical methods (Tsoukias and George, Pietropaoli, Condorelli, Högman and Meriläinen, and Silkoff) to estimate alveolar and bronchial NO parameters Logistic least absolute shrinkage and selection operator (lasso) regression, chi-squared test or Fisher’s exact, Kolmogorov–Smirnov test to assess normality, t-test or analysis of variance (ANOVA), Wilcoxon–Mann–Whitney test or Kruskal–Wallis test” are related to the all three first working groups (Lamote K., 2017 [87], Karvonen T., 2017 [93] and Lamote K., 2016 [87]) or to only the first one.  Please, make table 1 clearer.

Response of reviewers: The authors reviewed the Table according to the referee’s suggestion.

Reviewer 3 Report

This manuscript presents a systemic review of the strengths and preliminary results of breath  

 biomarkers of  Malignant pleural mesothelioma (MPM).  The identification of diagnostic VOCs pattern in breath samples  and the statistical   data analysis provided good diagnostic power.  However,  due to the   limited number of individuals included in each cohort study, the promising results and diagnostic accuracy, and conclusions cannot be used for clinical diagnosis.The topic is very important because of the challenges of diagnosis of Malignant pleural mesothelioma. The manuscript can be improved by including the following comments.

1.      Table 1. Add a column of sample information. For example, exhaled breath samples or EBC. Add another column of reported  HC and MPM patients biomarker concentrations ranges in exhaled breath.

2.      Table 1 should be mentioned in section 2.1 so that readers can check it sooner.

3.      In Section 2,  please include/discuss cancer stage information of MPM and biomarker concentrations in exhaled breath related to cancer stage.

4.      There are some errors in the manuscript

Page 2, Line 43,  remove lo in “due to the lo very long latency…”

There is no number and title of  Section 3.  There should be a title such as Methodology of the review.

Author Response

Dear reviewer,

we reviewed the paper according to your relevant and useful  suggestions. Moreover, the manuscript has been thoroughly revised in terms of content adding further information and for grammar and spelling by a mother tongue English scientist.

Please read through the manuscript. We answered, point by point, to your comments and reviewed the paper hoping that it will be more acceptable for publication.

This manuscript presents a systemic review of the strengths and preliminary results of breath   biomarkers of  Malignant pleural mesothelioma (MPM).  The identification of diagnostic VOCs pattern in breath samples  and the statistical   data analysis provided good diagnostic power.  However,  due to the   limited number of individuals included in each cohort study, the promising results and diagnostic accuracy, and conclusions cannot be used for clinical diagnosis. The topic is very important because of the challenges of diagnosis of Malignant pleural mesothelioma. The manuscript can be improved by including the following comments.

1.      Table 1. Add a column of sample information. For example, exhaled breath samples or EBC. Add another column of reported  HC and MPM patients biomarker concentrations ranges in exhaled breath.

Response of reviewer: The authors reviewed the Table according to the referee’s suggestion.

2.      Table 1 should be mentioned in section 2.1 so that readers can check it sooner.

Response of reviewer: The authors reviewed the Table according to the referee’s suggestion and mentioned it in section 2.1.

3.      In Section 2,  please include/discuss cancer stage information of MPM and biomarker concentrations in exhaled breath related to cancer stage.

Response of reviewer: The authors reviewed the Table according to the referee’s suggestion.

4.      There are some errors in the manuscript

Response of reviewer: The authors deeply checked the manuscript and reviewed the text.

Page 2, Line 43,  remove lo in “due to the lo very long latency…”

Response of reviewer: Done

There is no number and title of  Section 3.  There should be a title such as Methodology of the review.

Response to reviewer: The authors thank the reviewer and named the section.

Round 2

Reviewer 1 Report

Thank you.

The extensive changes have improved the paper.

Please refer to non-invasive (you have said not-invasive)

The paper detecion of liver disease by Khalid et al Metabolomics 2013 Should be added to the 2 in J Breath Research